# Thermal Conductivity Enhancement of Metal Oxide Nanofluids: A Critical Review

**DOI:** 10.3390/nano13030597

**Published:** 2023-02-02

**Authors:** Humaira Yasmin, Solomon O. Giwa, Saima Noor, Mohsen Sharifpur

**Affiliations:** 1Department of Basic Sciences, Preparatory Year Deanship, King Faisal University, Al-Ahsa 31982, Saudi Arabia; 2Department of Mechanical Engineering, Olabisi Onabanjo University, Ago-Iwoye P.M.B. 2002, Nigeria; 3Department of Mechanical and Aeronautical Engineering, University of Pretoria, Pretoria 0002, South Africa; 4Department of Medical Research, China Medical University Hospital, China Medical University, Taichung 404, Taiwan

**Keywords:** enhancement, metal oxides, nanofluids, nanoparticles, thermal conductivity

## Abstract

Advancements in technology related to energy systems, such as heat exchangers, electronics, and batteries, are associated with the generation of high heat fluxes which requires appropriate thermal management. Presently, conventional thermal fluids have found limited application owing to low thermal conductivity (TC). The need for more efficient fluids has become apparent leading to the development of nanofluids as advanced thermal fluids. Nanofluid synthesis by suspending nano-size materials into conventional thermal fluids to improve thermal properties has been extensively studied. TC is a pivotal property to the utilization of nanofluids in various applications as it is strongly related to improved efficiency and thermal performance. Numerous studies have been conducted on the TC of nanofluids using diverse nanoparticles and base fluids. Different values of TC enhancement have been recorded which depend on various factors, such as nanoparticles size, shape and type, base fluid and surfactant type, temperature, etc. This paper attempts to conduct a state-of-the-art review of the TC enhancement of metal oxide nanofluids owing to the wide attention, chemical stability, low density, and oxidation resistance associated with this type of nanofluid. TC and TC enhancements of metal oxide nanofluids are presented and discussed herein. The influence of several parameters (temperature, volume/weight concentration, nano-size, sonication, shape, surfactants, base fluids, alignment, TC measurement techniques, and mixing ratio (for hybrid nanofluid)) on the TC of metal oil nanofluids have been reviewed. This paper serves as a frontier in the review of the effect of alignment, electric field, and green nanofluid on TC. In addition, the mechanisms/physics behind TC enhancement and techniques for TC measurement have been discussed. Results show that the TC enhancement of metal oxide nanofluids is affected by the aforementioned parameters with temperature and nanoparticle concentration contributing the most. TC of these nanofluids is observed to be actively enhanced using electric and magnetic fields with the former requiring more intense studies. The formulation of green nanofluids and base fluids as sustainable and future thermal fluids is recommended.

## 1. Introduction

To enhance heat dissipation and efficiency of thermal systems, an improvement in thermal fluids is necessary, especially when different heat-enhancing methods have been exhausted and presently reaching their practicable limits [1,2]. This challenge ignited the engineering of superior thermal fluids with higher TC compared with the conventional thermal fluids (engine oil (EO), water, and ethylene glycol (EG)) as pioneered by Maxwell, Ahuja, and Masuda [1,3,4]. The first two researchers intended to enhance the thermal conductivity (TC) of conventional thermal fluids by the suspension of micro-scaled particles of metals and non-metals known to have higher TC than the conventional base fluids. However, the idea was plagued with erosion, clogging, sedimentation, and considerable pressure drop, which was resolved by the work of Masuda, which engaged nano-scaled particles of TiO_2_, Al_2_O_3_, and SiO_2_ suspended in water. This particular study gave birth to new thermal fluids coined “nanofluid”, which has and still is receiving global attention in the field of nanoscience and thermal engineering. Nanofluids are engineered colloids formulated by the suspension of nanoparticles of metals, metal carbides, ceramics, metal oxides, metal nitrides, and carbon (nanotubes/graphene sheets) with less than 100 nm in different conventional thermal fluids.

The subject of the thermal properties and performance of these innovative thermal fluids–nanofluids in various applications has been extensively studied by the research community [5,6,7,8,9,10,11,12,13,14,15,16,17,18,19,20]. It is worth mentioning that research progress on nanofluids led to the emergence of “hybrid nanofluids”, which was based on the initial concept of improving the TC of conventional thermal fluids. In this case, two dissimilar nanoparticles (with different physical and chemical properties leading to different TC) were synergized for improved TC, and the duo was suspended in water to formulate a hybrid nanofluid [21]. Further studies based on these pioneering works have been conducted concerning the thermal properties and performance of hybrid nanofluids [22,23,24,25,26,27,28]. For both mono-particle and hybrid nanofluids, TC has been the pioneering and focal thermal property. Other thermal properties of nanofluids (mono-particle and hybrid) studied are viscosity, specific heat capacity, density, surface tension, and electrical conductivity.

Nanofluid’s TC remains the most studied and important thermal property as it strongly correlates with the heat transfer performance and thermal efficiency of energy devices and systems [29]. The TC of nanofluids is related to various parameters, such as nanoparticle characteristics (size, concentration, TC, and shape) [30,31]; temperature [30]; stability, preparation, and sonication characteristics [32,33]; surfactant presence and quantity [30,31]; base fluid characteristics [34]; TC measurement methods [35]; and alignment [34,36] and marked by discrepancies. The suspension of different types of nanoparticles in various base fluids has been reported to lead to varying degrees of TC enhancement. Classical studies on the TC of mono-particle nanofluids demonstrated that Masuda and co-workers [37] published a 30% TC improvement when a Al_2_O_3_/water nanofluid with a volume fraction of 4.3% was measured. Choi and Eastman [38] revealed a 3.5-times augmentation of the TC of Cu/water nanofluid at a volume fraction of 20%. Contemporary studies in this regard reported TC enhancement by 30% (DW-based) and 31% (EG-based) for Al_2_O_3_ nanofluids with a concentration of 2 vol% at 70 °C [39]. A 54.6% TC improvement was recorded using graphene nanosheet/EG nanofluid at a weight fraction of 0.5 wt% and 70 °C [40]. Prado et al. [41] studied the TC of MgO/n-tetradecane nanofluids at varying temperatures and mass fractions. TC improvement of 4.2–17% was observed as TC enhanced with mass fraction and diminished with temperature.

For the hybrid nanofluids, studies reported higher TC compared to those of mono-particle nanofluids [21,42,43,44]. Classical studies showed TC enhancement of 50–150% for the suspension of Al_2_Cu and Ag_2_Al in EG and water with concentrations of 0.2–1.5 vol% [21], while 27.8% improvement was recorded for 0.05 wt% MWCNT + 0.02 wt% Fe_2_O_3_/water nanofluid [45]. For contemporary investigations of the TC of hybrid nanofluids, maximum enhancements of 28.5% [27] and 128.4% [46] were reported for MWCNT + Fe_3_O_4_/water (at 0.3% particle loading and 60 °C) and MWCNT-CuO/therminol55 nanofluids (at 0.08 wt% concentration and 80 °C), respectively. Recent work showed that the TC of W-bio glycol (60:40) based SiO_2_-Al_2_O_3_ nanofluids (with 0.5 vol% concentration, different mixing ratios, and temperatures) was higher than those of SiO_2_ and Al_2_O_3_ nanofluids [47]. A maximum TC enhancement of 21.2% was published for the SiO_2_-Al_2_O_3_ nanofluids with a mixing ratio of 70:30 at 70 °C. The TC of water-based CuO and CuO + Cu nanofluids with a weight fraction of 2 g/L under increasing temperature [44]. CuO/water nanofluid has a TC range of 0.615–0.712 W/m K whereas CuO + Cu nanofluids exhibited a TC range of 0.629–0.779 W/m K, all at the temperature range of 25–50 °C. Maximum TC was recorded for 2.5 g CuO + 1.5 g Cu nanofluid.

Progress in nanofluid research demonstrated the use of either green nanoparticles or green base fluids for the formulation of green nanofluids [48,49,50,51,52]. TC improvement of 16.1% was determined for EG-DIW (50:50)-based palm kernel nanofluid (at 0.5 vol% and 60 °C) [48]. With an EG-based hybrid nanofluid formulated using fruit bunch and GO nanoparticles, TC was recorded to be improved [50]. Bioglycol-based Al_2_O_3_ nanofluids were reported to yield higher TC compared with the use of conventional base fluids [51]. The deployment of a magnetic field to promote the TC of nanofluids has also been published [53,54,55].

The huge number of literature available in the public domain concerning studies on the measurement and enhancement of TC via the deployment of nanofluids (mono-particle and hybrid) has necessitated an updated review of this subject. Owing to the importance of TC to nanofluid applications, such as heat transfer and efficiency in thermal systems and devices, this present paper has focused on the TC of metal oxide nanofluid as they have attracted more attention, subject to their chemical stability, low density, and oxidation resistance. The effect of temperature, volume/weight concentration, nano-size, sonication, shape, surfactants, base fluids, alignment, TC measurement techniques, and mixing ratio (for hybrid nanofluid) on the TC of metal oxide nanofluids has been compiled and discussed. The measurement techniques and mechanisms related to the TC of nanofluids have also been presented. A special feature of this work is the deployment of the magnetic and electric fields, and green metal oxide nanofluid as active and passive techniques, respectively, to improve the TC of metal oxide nanofluids, which is lacking in previous review studies. In addition, the influence of alignment on TC of metal oxide nanofluid is discussed as an under-reported parameter affecting nanofluid TC. This work aims to present a holistic document on the TC of metal oxide nanofluids which will immensely contribute to nanofluid research and benefit the research community. The trend of nanofluid TC from 1999 to 2022 is provided in Figure 1 and in Table 1.

This paper is divided into eight sections. The first section is the introduction of this paper which entails the motivation and objectives of this work. Techniques deployed in the measurement of TC of nanofluids are highlighted and discussed in Section 2. The TC enhancement mechanisms and contributing factors are presented and deliberated in Section 3 and Section 4, respectively. TC enhancement techniques and green nanofluid development studies have been compiled and discussed in Section 5 and Section 6, respectively. The challenge and future research, and conclusion are presented in Sections seven and eight, respectively.

## 2. TC Measurement Methods

The need to measure the properties of materials, especially thermal fluids, is very important and strongly related to the application of the same. The development of an appropriate measuring technique is seriously connected to the state, type, physical, and chemical composition of the material, the physics and mechanisms behind the measured property, etc. Additionally, reliability and correctness of the measured properties is important. TC is an indicator of the heat transfer ability of a material. In the quest to measure the TC of nanofluids via an experimental approach by researchers, various techniques have been developed and reported in the literature. These techniques are generally classified as steady-state, transient, and thermal comparator techniques [31]. The steady-state method is sub-divided into parallel-plate and cylindrical cell methods while the transient technique is further classified as 3ω, temperature oscillation, transient hot-wire, laser flash, and thermal constant analyzer techniques. Figure 2 illustrates the classification of nanofluid TC measurement techniques. Additionally, the transient hot-wire method is further categorized as transient short hot-wire and liquid metal transient hot-wire methods [35]. TC is mathematically expressed based on the Fourier law, as given in Equation (1) [29]:(1)$$\kappa =\frac{Q.dx}{A.dT}$$
where *κ* = TC W/(m K); *Q* = quantity of heat passing through a cross-sectional area *A* (m^2^) which leads to a temperature difference; *dT*/*dx* = temperature gradient over a distance of *dx* (K/m).

Of the above-mentioned TC measuring techniques, the transient hot wire and the thermal constants analyzer are the most used techniques for measuring the TC of nanofluids [29]. These techniques fall under the broad category of transient technique that is characterized by a local temperature difference varying as dependent on time. The design and construction of a high-precision and accurate thermal constants analyzer and transient hot-wire measurement device is challenging. However, the use of the transient hot wire and the thermal constants analyzer techniques for nanofluid TC measurement are found to be associated with systematic errors due to natural convection currents and the capacitance effect within the measured nanofluids [71]. At higher temperatures and using the transient hot wire technique, the initiation of natural convection is reported to lead to higher TC than obtained when the steady-state method is used [31]. Other sources of systematic errors in nanofluid TC measurement include dependence and high sensitivity to samples’ initial conditions, nanofluid stability difficulty, nanofluid concentration, and specific heat of nanofluid components [72].

Although the devices developed via the mentioned techniques have different degrees of sophistication, they have their merits and demerits. The deployment of different techniques for measuring nanofluid TC is reported to be marked by inconsistency in measured values [71]. The use of the transient hot wire approach was observed to measure a higher TC for water-based Ag and Al_2_O_3_ nanofluids compared to the use of the laser flash technique for the same purpose [73]. This was due to the demonstration of more collision flux with the wall by the nanoparticles using the transient hot wall approach. Additionally, the engagement of the transient hot wire technique to measure the TC of Al_2_O_3_/water nanofluid was found to yield higher enhancement (16.5%) than that conducted using the laser flash approach (4.95%) [74]. A comparison study conducted on the use of different TC techniques of transient hot wire, laser flash, and thermal constant analyzer showed that the transient hot wire produced the best results in terms of repeatability and precision [75]. Figure 3 presents a comparison of measurement techniques for the TC of Al_2_O_3_ nanofluids. It is worth mentioning that all the TC techniques are well-developed as they have undergone improvement over time. For further studies on the design, development and evolution, operation, and uncertainty analysis of all the techniques, please see the literature [29,71].

## 3. TC Enhancement Mechanisms

Early studies on nanofluid heat conduction show enhancement of up to 40% in TC with a nanoparticle concentration of less than 5% [5,76,77]. Different mechanisms also have been proposed for this anomalous enhancement [75,78] which include static mechanisms—nanolayering, aggregation and percolation, interface thermal resistance, fractal geometry, and dynamic mechanisms—Brownian motion, ballistic nature of nanoparticles, and nanoscale convection.

### 3.1. Brownian Motion of the Nanoparticles

Generally, there are three types of motion regarding the movement of nanoparticles in nanofluids. These are Brownian, thermophoretic, and osmophoretic motions which are due to force, temperature difference, and concentration gradient, respectively. The anomalous improvement in nanofluid TC was linked to the effect of Brownian motion which involves the random motion of nanoparticles in the base fluid owing to continuous bombardment of the particles and base fluid molecules. During Brownian motion, first there is thermal transport due to particle-particle interaction leading to improved TC as nanoparticles have a high volume-to-area ratio. The second is heat transfer via micro-convection due to particle-fluid interaction [30]. Brownian motion has the most effect on nanofluid TC than thermophoretic and osmophoretic motions [79]. However, the insignificance of Brownian motion to TC improvement of nanofluid has been reported [71].

### 3.2. Nanolayer Effect

The nanolayer is the ordered solid-fluid interface formed owing to the strong particle-fluid force of interaction. The TC of the nanolayer is reported to be more than that of the bulk base fluid and lower than that of the nanoparticle [80]. It is said to function as a thermal bridge between the base fluid and the nanoparticle as the thickness of the nanolayer increases the concentration of nanoparticles in the base fluid causing TC enhancement of nanofluids [30,80]. The presence of a particle-fluid interface introduces an interfacial thermal resistance called “Kapitza resistance”, which serves as an obstacle to heat transfer and therefore reduces the overall TC within the system. Although, nanolayer thickness is of the order of a nanometer, but due to the high specific surface area of nanoparticles, the nanolayer effect becomes critical and plays a key role in heat transfer across the particle-fluid interface [71].

### 3.3. Nanoparticle Clustering

Nanoparticle constituent of nanofluid clusters as the distance between nanoparticles becomes smaller during the collision as the weak force of attraction (van der Waals) increases [71]. At high nanoparticle concentration, nanoparticle clustering possibility increased. The clustering of nanoparticles in nanofluids has been reported to improve nanofluid TC [81]. This is due to localized rich-particle portion development with lower thermal resistance to heat transfer compared to the less-particle portion. The formation of larger particle-free portions due to the settling of heavier aggregates lowers the TC. The cluster (particle-rich portion) of nanoparticles in the nanofluid contains more particles than the less-particle portion leading to a quicker transfer of heat [30].

### 3.4. Ballistic Nature of Nanoparticles

In solid and micro-scale, heat is transferred as phonons that are formed and propagated at random and dispersed by one another [71]. Heat is conducted in a solid material via the vibration of atoms jointly held together. The vibrating arrangement of atoms releases or losses energy in quantized form as a phonon. Thus, a phonon performs a key role in the TC of a material. In a hot region, a higher phonon density exists compared to that of a cold region, thus implying heat transfer is largely due to phonon diffusion subject to temperature gradient [30]. This is easily related to the Ballistic behaviour of nanoparticles as the size of nanoparticles is smaller than the atomic scale phenomenon of phonon heat transfer mechanisms. Higher ballistic phonon transport mechanisms are experienced in a nanofluid as the nanoparticle size reduces [82].

## 4. TC Enhancement Contributing Factors

### 4.1. Concentration

Factors contributing to the TC of nanofluids are provided in Figure 4. Nanofluid is formulated by the suspension of nanoparticles in a base fluid. Increasing the quantity of nanoparticles suspended in the base fluid will directly increase the concentration of the nanoparticles in the base fluid. The presence of nanoparticles is expected to enhance the TC of the formulated nanofluid as the existence of Brown motion and other mechanisms aid TC enhancement. Two-fold studies have been published concerning the effect of concentration on the TC of nanofluids. Classical studies for both mono-particle and hybrid nanofluids were conducted at room temperature to measure their TC while subsequent works measured TC under varying temperatures. The work of [37] revealed the TC enhancement of water-based TiO_2_ and Al_2_O_3_ nanofluids as the concentration increased, while no improvement in TC was observed for SiO_2_/water nanofluid. An increase in volume concentration (1–5 vol%) was observed to augment the TC of water and EG-based CuO and Al_2_O_3_ nanofluids with a maximum enhancement of 20% for CuO/EG nanofluids [76]. The TC of Al_2_O_3_ and CuO nanoparticles suspended in vacuum pump oil, EG, engine oil, and DIW at room temperature demonstrated augmentation with concentration with a maximum enhancement for Al_2_O_3_/EG (40% with 8 vol%) [69]. With a maximum enhancement of 1.44%, the TC of Al_2_O_3_/water nanofluids showed an improvement as the volume fraction increased from 0.01% to 0.3% [63]. An enhancement of the TC of 0.2 vol% TiO_2_/water nanofluid measured at room temperature was reported [61].

Under increasing volume concentration of TiO_2_/EG, the TC was observed to increase with an improvement of 18% [83]. The TC of water-based CuO and Al_2_O_3_ nanofluids was found to enhance with an increase in concentration from 1 vol% to 4 vol% at room temperature [62]. Enhancements of 2–9.4% and 6.5–14% at room temperature were recorded for Al_2_O_3_ and CuO nanofluids at 1 vol% and 4 vol%, respectively. The TC of Fe_3_O_4_/DIW nanofluids showed an increase as the volume concentration increased from 0.2–2 vol% with a maximum enhancement of 48% [64]. An increase in volume concentration of Fe_3_O_4_/kerosene nanofluids (0–1 vol%) was observed to directly enhance the TC [65]. The TC of MgO/glycerol nanofluid was enhanced by 19% as the volume fraction increased from 0.5% to 4% [14]. At 30 °C, the TC of ZnO/EG nanofluid was enhanced by 40% when the nanofluid concentration was increased from 0.5 vol% to 3.75 vol% [84]. The TC of EG-based αFe_2_O_3_ and Fe_3_O_4_ nanofluids at a volume fraction of 0.69% was observed to be augmented by 15% and 11%, respectively [85]. The TC of Al_2_O_3_/DIW nanofluids was enhanced by 15% for a weight fraction of 0.8% at room temperature [86].

With the use of Al_2_O_3_/bioglycol nanofluid, an increase in volume concentration was noticed to yield a maximum TC enhancement of 24% using a concentration of 2 vol% for 40 bioglycol-60 water-based nanofluid [67]. The formulation of bioglycol-based Al_2_O_3_ nanofluids was observed to afford improved TC (17%) for the concentration of 1 vol% and at 30 °C [51]. Under increasing volume fraction, the TC ratio of Ag-MgO (50:50)/DW nanofluid was observed to enhance [87]. The TC of Cu-Al_2_O_3_ (10:90)/DIW nanofluid was found to be augmented by 1.47–12.11 as the concentration rose from 0.1 vol% to 2 vol% [60]. A maximum TC of 27.8% was recorded for MWCNT-Fe_2_O_3_/water nanofluid with 0.05 wt%:0.02 wt% concentration [45]. Generally, increasing the concentration of nanoparticles has been found to augment the TC of nanofluids. The TC of MWCNT-CuO/therminol55 nanofluids was observed to enhance as the concentration rose from 0.005 wt% to 0.08 wt% with the highest augmentation of 128.4% [46].

### 4.2. Temperature

Temperature is an important property that influences the TC of nanofluids. As the temperature of nanofluids rises, an increase in the kinetic energy of base fluid molecules and nanoparticles occurs. This leads to intensified micro-convention, Brownian motion, and bombardment between particle–molecule and particle–particle resulting in increased TC of nanofluids. The TC ratio of TiO_2_/EG and Al_2_O_3_/DIW increased as the temperature rose from 20–60 °C with TC enhancement of 18% (5 vol% and 60 °C) and 12% (1 vol% and 60 °C), respectively [61]. The TC of water-based CuO and Al_2_O_3_ nanofluids was found to enhance with an increase in concentration from 1 vol% to 4 vol% at room temperature [62]. At 4 vol% and 51 °C, TC enhancements of 9.4–24.3% and 14–36% were observed for Al_2_O_3_ and CuO nanofluids, respectively. As the temperature of Fe_3_O_4_/DIW nanofluids increased from 20 °C to 60 °C, the TC was enhanced from 8.4–17% and 25% to 48% of Fe_3_O_4_/DIW nanofluids for the concentration of 0.2 vol% and 2 vol%, respectively [64].

An increase in temperature from 10–60 °C was observed to enhance the TC of Fe_3_O_4_/kerosene nanofluids by a maximum value of 34% [65]. At a concentration of 2 vol%, temperature increase from 10 °C to 70 °C for EG and DW-based Al_2_O_3_ nanofluids was found to enhance their TC by 31% and 30%, respectively [39]. At 0.3 wt%, the TC of SiO_2_ nanofluid was reported to diminish with an increase in EG content of the EG-water base fluid and enhance with a temperature rise from 25–45 °C [66]. Recently, the TC of WO_3_/EG nanofluids was found to be improved by 32.4% as the temperature increased from 5–65 °C for a mass fraction of 1.5 wt% [88]. Under increasing temperature (10–50 °C) and concentration (1–7 vol%), the TC of TiO_2_/EG was enhanced by 2.7–19.52% [68].

By increasing the temperature and concentration, the TC of DW and DW-EG based ND-Fe_3_O_4_ (72:28) nanofluids was improved by 17.8% and 13.4–14.6%, respectively when concentration was 0.2 vol% and at 60 °C [56]. It was observed that as EG content increased the enhancement was increased to 60 °C, respectively. With the DW and DW-EG-based ND-Fe_3_O_4_ (67:33) nanofluids subjected to increasing volume concentration (0.05–0.15 wt%) and temperature (20–60 °C), the TC was enhanced by 2.1–15.7% [18]. Using TiO_2_–SiO_2_/EG nanofluids, the effect of increasing volume concentration and temperature on the TC revealed a peak improvement of 22.1% with 3 vol% concentration and at 70 °C [59]. Under increasing temperature (20–100 °C) and weight concentration (0.005–0.08 wt%), the TC of MWCNT-CuO/therminol55 nanofluids was accessed leading to enhancement of 30.6–128.4% [46]. This remarkable enhancement was attributed to the ultrathin nanolayer between the nanoparticle-base fluid interface. The impact of temperature (20–40 °C) on the TC of DIW-based Fe_2_O_3_-MWCNT (80:20) and Fe_2_O_3_-Al_2_O_3_ (75:25) nanofluids with concentrations of 0–0.4 vol% and 0–0.3 vol%, respectively, was accessed and found to be enhanced by 3.84–14.17% and 0.58–3.32%, respectively [89,90].

Contrary to the above results, an increase in temperature from 20 °C to 45 °C for MgO/glycerol nanofluids with volume fractions of 0.5–4% was observed not to affect the TC despite the improvement of the TC of glycerol through the suspension of MgO nanoparticles in it [14]. Additionally, the TC of EG-based Fe_2_O_3_ and Fe_3_O_4_ nanofluids was found to be independent of temperature rise from 10 °C to 50 °C with the enhancement of 15% and 11%, respectively, at a volume fraction of 0.69% [85]. Owing to the effect of nanoparticle clustering as a result of nanofluid stability which can negate Brownian motion, the temperature change of nanofluids may not always favor the enhancement of TC of nanofluids. A different result was demonstrated when the TC of water and EG-based Fe_3_O_4_ nanofluids were measured [91]. It was reported that the TC diminished with an increase in concentration and temperature, which was related to the combined effect of interfacial thermal resistance and surfactant layer charge. The effect of temperature and concentration on the TC of nanofluid is illustrated in Figure 5.

### 4.3. Nanoparticle Size

The size of a particle is a unique characteristic for classification and identification attributable to some important properties. In the case of nanoparticles, the particle size is traceable to the name and the distinct properties (thermal and convective) connected to its use for nanofluid formulation. The use of different sizes of nanoparticles to formulate nanofluids is critical to the TC and stability of the resulting nanofluids. The nanofluid scientific community remains divided on the influence of nanoparticle size on the TC of nanofluids. This is marked by different scientific opinions behind their perceived results. An increment in nanoparticle size has been observed to either enhance or reduce the TC of nanofluids [31]. A school of thought based on experimental works reported a direct relationship between nanoparticle size and TC. The increase in nanolayer thickness, nano-clustering, surface area, nano-convection, and Brownian motion due to a reduction in nanoparticle size has been linked to the enhancement of nanofluid TC [31]. The influence of nanoparticle size on nanofluid TC is presented in Figure 6.

The measurement of the TC of water-based Al_2_O_3_ with nanoparticle sizes of 28 nm [69], 38 nm [76], and 13 nm [37] resulted in enhancements of 12%, 8%, and 20%, respectively. This showed an increase in the TC as the nanoparticle size reduced. The TC of water and EG-based ZnO (10–60 nm) and TiO_2_ (10–70 nm) nanofluids showed an improvement in this property as the concentration increased (1–3%), and the sizes of the nanoparticles for both nanofluids diminished [92]. The influence of micro-convection on the TC of 5.5 vol% Fe_2_O_3_/water nanofluids with nanoparticle sizes of 2.8 nm and 9.5 nm was studied [93]. Enhancement of TC by 5% and 25% was recorded with nanoparticle sizes of 2.8 nm and 9.5 nm, respectively, which is strongly related to micro-convection as a result of Brownian motion. Additionally, a trend of improvement in TC as nanoparticle size decreased was observed when EG and water-based Al_2_O_3_ nanofluids were examined for their thermal conductivities under changing temperature (20–50 °C), volume fraction (0.5–3%) and nanoparticle size (11–150 nm) [94]. Peak enhancement of 11–32% and 9.5–11% were reported with nanoparticle size of 11 nm, temperature of 50 °C, and volume fraction of 3% for Al_2_O_3_/water and Al_2_O_3_/EG nanofluids, respectively.

The impact of nanoparticle size (MgO—20 and 100 nm), temperature, and mixing ratio on the TC of DIW-based MgO and MgO–ZnO nanofluids (at 0.1 vol%) was conducted [43]. A decrease in the nanoparticle size of MgO was observed to intensity TC of MgO and MgO–ZnO nanofluids. At 25 °C and under changing nanoparticle size and volume concentration, the TC of water-based SiO_2_, TiO_2_, Al_2_O_3_, and ZrO_2_ nanofluids was measured [95]. It was observed that increasing nanoparticle size enhanced the TC ratio for all nanofluids. The study showed that subject to varying volume fraction and nanoparticle size (at ambient temperature), the TC of EG and water-based Al_2_O_3_ nanofluids enhanced as the nanoparticle size increased in the range of 2 nm to 50 nm [74]. This finding was linked to the phonon scattering at the particle-fluid interface. The influence of varying nanoparticle size (Al_2_O_3_—5 nm and 30 nm), temperature, mixing ratio (10:90–90:10), and volume concentration on the TC was examined [96]. A direct relationship was observed between the nanoparticle size and TC with a maximum augmentation of 45.1%.

The impact of volume fraction (0–5%) and nanoparticle size (spherical (15 nm) and rod-shaped (40 nm) on the TC of Ti_2_O/DIW nanofluids showed enhancement (33%—40 nm and 30%—15 nm) with an increase in nanoparticle size [97]. A study on the influence of different base fluids, volume fractions, specific surface areas, and nanoparticle sizes (12.2–302 nm) on the TC of Al_2_O_3_ nanofluids revealed that for all the nanofluids the TC augmented with a rise in nanoparticle size from 12.2 to 60.5 nm while the reverse was reported when the nanoparticle size increased from 60.5–302 nm [98]. Additionally, the specific surface area was noticed to increase as the nanoparticle size increased. The obtained results were connected to the phonon mean free path. When the nanoparticle size is larger than the phonon means free path, TC enhancement occurs as the specific surface area increases for improved particle-fluid interaction. With a smaller or equal nanoparticle size to that of the phonon means free path, a reduction in TC is observed as a result of phonon scattering at the particle interface. In addition, excessive particle clustering especially for small-size nanoparticles has been reported to lead to a reduction in nanofluid TC as nanoparticle size was reduced [99].

### 4.4. Base Fluid Characteristics and Alignment

The characteristics (such as polarity, viscosity, and hydrogen bonding) of base fluids utilized in the formulation and application of nanofluids are very important [34]. Suspending different nanoparticles with their peculiar chemical and physical properties in various base fluids with dissimilar characteristics is a complex exercise and a good understanding of these two basic materials and their peculiarities is key to the choice of these materials, experimental results, and nanofluid applications. The thermal properties of nanofluids (for example TC) are strongly connected to the existence and the degree of thermal interfacial resistance of the base fluid molecules and the suspended nanoparticles. Metal oxide-based nanoparticles have been reported to be well-dispersed in highly polarized base fluids [34]. The influence of EG and EG-W as base fluids on the TC of 4 vol% SiC nanofluids with difference nano-sizes was studied [100]. Using EG-W-based SiC nanofluid was noticed to exhibit higher TC enhancement than W-based SiC nanofluid. The outcome was connected to the reduction in the interfacial thermal resistance value of the EG-W compared to W.

A deeper understanding of the effect of base fluid characteristics (polarity, hydrogen bonding, and viscosity) on the TC of Fe_2_O_3_ nanofluids with and without a magnetic field was provided by the work of Christensen et al. [34]. Fe_2_O_3_ nanoparticles were suspended in twelve different solvents with diverse characteristics. In the presence and absence of magnetic effect, the suspension and alignment of Fe_2_O_3_ nanoparticles were enhanced as base fluids with a single OH group exhibiting inter-molecule hydrogen bonding caused lower viscosity and higher polarity which improved the TC of the corresponding nanofluids. Exposure of the nanofluids to a magnetic field increased nanoparticle alignment leading to an increased TC enhancement. In addition, Hong et al. [101] demonstrated that the alignment of SWNT-Fe_2_O_3_ nanofluid (using NaDSSB as a surfactant) caused TC improvement, especially under the influence of a magnetic field. TC was enhanced to the maximum (1.36 W/m K) when the nanofluid was exposed to the magnetic field for 30 s. A change in nanoparticles (from Fe_2_O_3_ to NiO) and surfactant (from NaDSSB to CTAB) was reported to corroborate the effect of alignment on TC.

A study on the influence of alignment and base fluids (water, EG, and water-NaDDBS) on the TC of Fe_2_O_3_ and CuO nanofluids (at 0.4 vol%) was conducted [102]. The Fe_2_O_3_ nanofluids exhibited higher TC values compared to CuO nanofluids as the particles of the Fe_2_O_3_ nanofluids were observed to align without a magnetic field. Water-based nanofluids have the highest TC value, followed by water-NaDDBS, then EG. In the presence of a magnetic field, the TC of Fe_2_O_3_/water and Fe_2_O_3_/water-NaDDBS nanofluids was enhanced while that of EG-based Fe_2_O_3_ nanofluid diminished. The influence of alignment of 0.017 wt% MgO-SWCT + 0.17 wt% NaDDBS nanofluid on TC was investigated [103]. Highest TC (0.92 W/m K) was recorded when SWCT nanoparticles were aligned on MgO under the influence of a magnetic field. This was around a 35% improvement over a case of no magnetic field effect. Sundar et al. [104] studied the effect of DIW and EG on the TC of GO-Co_3_O_4_ nanofluids. TC improvement of 11.85% and 19.14% was obtained using EG and DIW, respectively at 0.2 vol% and 60 °C.

### 4.5. pH

Different base fluids have different pH values. The suspension of different nanoparticles in base fluids alters the base fluids’ surface charge, and thus the pH, which is a function of the stability of the corresponding nanofluids. Modification of nanofluid surface charge via the pH to improve nanofluid stability affects the TC of the nanofluid [105]. Hydroxyl group formation is experienced when metal oxide nanoparticles are suspended in water. The nanoparticle surface charge polarity is linked to the isoelectric point of the solid phase and the base fluid pH. The pH of a cylinder boehmite Al_2_O_3_/EG nanofluid with 5 vol% concentration was modified from 2.54 to 4.10 to study its effect on thermal conductivity [106]. A slight improvement in thermal conductivity was observed as the pH increased.

A study on the influence of pH on the TC of water, EG, and water-NaDDBS-based Fe_2_O_3_ and CuO nanofluids (at 0.4 vol%) was conducted [102]. Although the pH of CuO nanofluids was higher than Fe_2_O_3_ nanofluids with EG demonstrating the highest pH of all the base fluids, changes in pH of Fe_2_O_3_ nanofluids were observed to appreciably improve the TC away from the iso-electric point. With DIW-based ZrO_2_ and TiO_2_ nanofluids, the impact of pH on the TC was examined [107]. Near the iso-electric point (pH = 6.2), the TC was significantly enhanced when altering the pH from 4 to 10. The effect of pH of 0.017 wt% MgO-SWCT + 0.17 wt% NaDDBS nanofluid on TC was examined [103]. Increasing the pH of the nanofluid from 7 to 11.5 was found to reduce the TC. In addition, the effect of altering the pH of αAl_2_O_3_/water nanofluids (with a volume fraction of 1.8 to 5) on TC was investigated [98]. An increase in the pH of the nanofluids was noticed to improve the TC. The farther the pH from the isoelectric point (9.2), the higher the TC. Stability and pH influence on the TC of 0.5 wt% DIW-based Al_2_O_3_ and CuO nanofluids were investigated [108]. Stable nanofluids were achieved at a pH of 8 (Al_2_O_3_) and 9.5 (CuO). Below these values, the TC enhancement (15% for Al_2_O_3_ and 18% for CuO) of these nanofluids was observed, and above these pH values, depreciation in TC was reported. In another study, the effect of pH on the TC of CeO_2_-MWCNT (80:20)/water nanofluids (0.25–1.5 vol%) formulated using six different surfactants under increasing sonication duration and surfactant-particle ratio was conducted [109]. At a surfactant-particle ratio of 3:2, sonication time of 90 min, and using CTAB, the best stability was exhibited at a pH of 9.5. A linear increment in TC was observed as the pH rose from 8 to 9.5 with a reduction in TC noticed as the pH increased beyond 9.5. This trend was observed for all the studied surfactants. Increasing the pH from 8 to 9.5 caused TC improvement from 7.2% to 13.1%.

### 4.6. Surfactants

The deployment of surfactants in the nanofluid formulation is aimed at improving the stability of nanofluids and preventing their segregation and settlement. Nanofluid TC is affected by nanofluid stability status. At low surfactant concentration, nanofluid TC is increased while at high concentration, TC is reduced [105]. Stability, surfactant (SDBS) weight fraction, and pH influence on the TC of 0.05 wt% DIW-based Al_2_O_3_ and CuO nanofluids were investigated. At pH of 8 (Al_2_O_3_) and 9.5 (CuO), stable nanofluids were obtained. At optimal SDBS weight fractions, optimal stability in terms of zeta potential and particle size was achieved. The influence of different surfactants (CTAB, SDS, and SDBS) on the TC of GnP and GnP-TiO_2_ nanofluids under varying sonication duration was investigated [110]. SDS and CTAB (at 30 min sonication) were the best surfactants for mono and hybrid nanofluid formulation with maximum TC improvement of 23.7% and 21.6%, respectively, at 60 °C and concentration of 0.1 wt%. Arasu et al. [111] studied the effect of different surfactants (SDS and SDBS) on the stability and TC of TiO_2_-Ag/water nanofluids (0.1–0.7 wt%). Results revealed the change in TC improvement with the use of different surfactants. SDS produced more stable nanofluids that exhibited a higher TC enhancement of 29.6% compared to SDBS with 2.1% TC improvement.

The effect of four different surfactants (acetic acid, SDS, CTAB, and SDBS) on the TC of TiO_2_/water nanofluids was studied [112]. Most stable nanofluids were formulated using SDS and CTAB. Highest improvement (5.8% at 60 °C and volume fraction of 1%) in TC was recorded with the use of SDS for the nanofluid formulation. Under varying sonication duration and surfactant-particle ratio, the influence of six different surfactants on the TC and stability of CeO_2_-MWCNT (80:20)/water nanofluids was examined [109]. Peak stability as measured by the zeta potential was attained at the optimum surfactant-nanoparticle ratio, and sonication time of 3:2 and 90 min, respectively. Maximum TC ratio was achieved using a CTAB-nanoparticle ratio of 3:2, volume concentration of 0.75%, 90 min sonication, and at 30 °C.

The above studies demonstrated that since surfactants have different characteristics, a specific surfactant does not apply to all types of nanoparticles and base fluids. The compactivity of base fluids and nanoparticles is crucial to the selection of surfactants for nanofluid formulation as the stability of nanofluids impacts their thermal properties and performance.

### 4.7. Mixing Ratio (Hybrid Nanofluids)

The influence of varying mixing ratio (20:80–80:20) and temperature (30–80 °C) on the TC of 1 vol% TiO_2_–SiO_2_/water–EG (60:40) nanofluid was found to improve this thermal property by 16% (peak) using mixing ratio of 20:80 at 80 °C [113]. The TC of 0.1 vol% MgO–ZnO/DIW nanofluids under the effect of changing mixing ratio and temperature was observed to be augmented by 15–22% using a mixing ratio of 40:60 (MgO–ZnO) at 50 °C [43]. The effect of the mixing ratio on the TC was observed to be higher than the temperature. The impact of different mixing ratios (30:70–70:30) on the TC of Al_2_O_3_–Ag/DW nanofluid under increasing volume fraction and temperature was conducted [114]. The TC was enhanced as the temperature and concentration increased with a peak enhancement of 23.6% using 0.1 vol% Al_2_O_3_–Ag/DW nanofluid having a mixing ratio of 50:50 and at 52 °C. The TC of water-based Fe_3_O_4_ + CNT nanofluids with varying mixing ratios (1:2, 1:1, and 2:1) was studied [115]. Maximum TC enhancement of 45.4% was recorded using 0.9 vol% Fe_3_O_4_ + 1.35 vol% CNT nanofluid.

The influence of mixing ratios (33.4:33.3:33.3, 50:25:25, 60:30:10, 25:50:25, and 25:25:50) of CuO:MgO:TiO_2_ nanoparticles on the TC of CuO–MgO–TiO_2_/water nanofluids under increasing volume concentration and the temperature was investigated [116]. Volume concentration and temperature rise were observed to augment TC. Maximum TC was achieved using 0.5 vol% CuO–MgO–TiO_2_/water nanofluid with a mixing ratio of 60:30:10 and at 60 °C. The TC of MgO–TiO2/DW nanofluids with mixing ratios (50:50, 80:20, 20:80, 60:40, and 40:60) was examined by varying the temperature and volume concentration [117]. As the temperature and concentration increase enhanced the TC, a peak enhancement of 21.8% was observed for 0.3 vol% MgO–TiO_2_ (80:20)/DW nanofluid at 60 °C. The impact of mixing ratio (75:25–25:75) on the TC of 0.1 vol% GNP–Al_2_O_3_ nanofluids under varying temperatures (20–40 °C) was examined. Peak enhancement of 1.83–3.42% was observed for GNP–alumina (75:25) at 40 °C [118]. The influence of varying nanoparticle size (Al_2_O_3_—5 nm and 30 nm), temperature (15–55 °C), mixing ratio (10:90–90:10), and volume concentration (0.025–0.5 vol%) on the TC was examined [96]. The use of a mixing ratio of 40:60 resulted in a maximum improvement of 45.1% attained at 0.5 vol% and 55 °C with a nanoparticle size of 30 nm. Figure 7 shows the impact of the mixing ratio on the TC of hybrid nanofluids.

### 4.8. Sonication Characteristics

The stability of nanofluid is very important to its application. Whether a surfactant is used or not to improve the stability of nanofluid, the deployment of the ultrasonication process is crucial to break up nanoparticles suspended in base fluids and aid even dispersion of nanoparticles into them. Sonication is to provide sufficient energy to overcome the interparticle attraction forces holding the nanoparticles together. The ultrasonication process is a complex exercise as it involves several variables such as sonication duration, amplitude, frequency, pulse time, probe depth, etc. Stability is known to affect the thermal and convection properties of nanofluids and this is strongly related to the sonication variables [32,119]. The influence of sonication characteristics on the TC of different nanofluids has been studied and contradictory results have been published in this regard in the scientific community [70,84,110,120]. Figure 8 illustrates the effect of sonication time on nanofluid TC.

#### 4.8.1. Mono Nanofluids

The influence of sonication time (2–8 h) on the stability and TC of DIW-based ZnO and CuO nanofluid (0.1 wt%) [121] was examined. Increasing sonication time was observed to affect TC and the need to optimize the TC relative to stability and sonication time was proposed. The effect of sonication time (20–60 min) on the stability and TC of WO_3_/EG nanofluids (0.005–5 wt%) was assessed [88]. For all the studied samples, the TC was augmented as the sonication time increased. The influence of MWCNT (0.01–1 *w*/*v*) and Fe_2_O_3_ (0.1–2 *w*/*v*) nanoparticles, pH (2–10), and EG-W (30–70 *v*/*v*) volume on the TC of MWCNT-Fe_2_O_3_ NFs was conducted [122]. Peak TC (0.534 W/mK) was reached at optimum values of 6.5 (pH), 1.67 *w*/*v* (Fe_2_O_3_), 44 *v*/*v* (EG), and 0.69 *w*/*v* (MWCNT) with MWCNT nanoparticles impacting TC the most. Under changing sonication time (30–150 min) and concentration (0.5–2 vol%), the TC of Al_2_O_3_/DW nanofluids was monitored [120]. Results showed that the TC improved as sonication time increased along with concentration to a certain point after which it reduced as sonication time increased further due to re-agglomeration. The highest TC enhancement of 4.6% and 16.1% for 1.5 vol% and 2 vol% concentration at sonication time of 90 min and 120 min, respectively, was recorded.

**Figure 8 nanomaterials-13-00597-f008:**
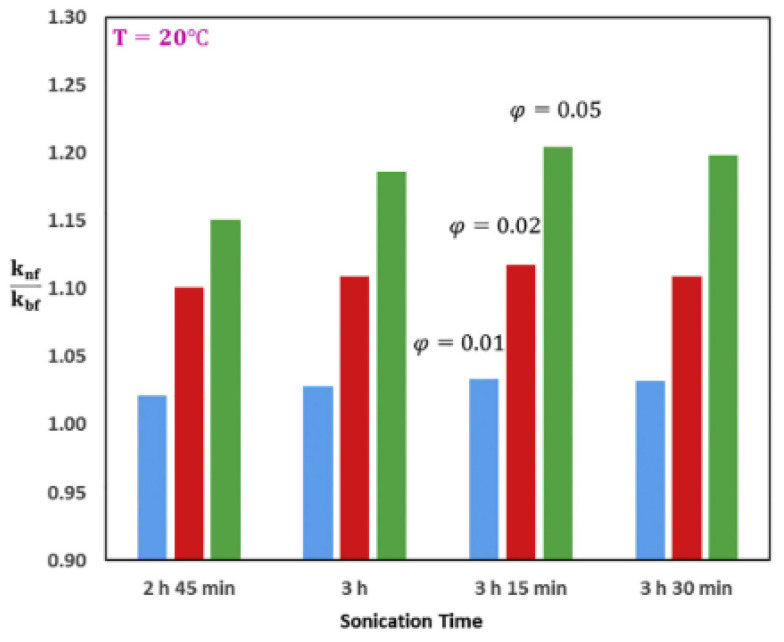
Effect of increasing sonication time and volume fraction on thermal conductivity ration of Al_2_O_3_/paraffin nanofluids at 20 °C [123].

The impact of sonication time (0.5–5 h) and energy (8.46–42.30 kJ) on the TC of 0.5 vol% Al_2_O_3_/DW nanofluid was investigated [124]. Increment in the sonication time was noticed to augment TC. By changing surfactant mass fraction (0.25–4 wt%), surfactants (SDS and PVP), concentration (0.1–2.5 vol%), nanoparticle (13 nm and 20 nm), and sonication time (0.25–2 h), the stability and TC of αAl_2_O_3_/DIW nanofluids were examined [70]. It was demonstrated that PVP provided better stability, but with lower TC. The TC remained constant when the mass fraction was >1.0 wt% but enhanced with sonication time (up to 1 h). Nanofluid with 13 nm particle yielded higher TC for both surfactants. The TC of γAl_2_O_3_/DIW nanofluids with varying volume concentration (1–3 vol%) and sonication duration (15–180 min) was examined [125]. It was demonstrated that the TC enhanced with sonication time.

Sonication time (4–100 h) influence on the TC and stability of ZnO/EG nanofluids (0.5–3.75 vol%) was conducted [84]. Increasing the sonication time from 4 h to 60 h was observed to cause TC improvement by 21–40% with the opposite noticed after a further increase in the sonication time. At 60 h sonication, TC remained constant with no sedimentation for 30 days for 1 vol% ZnO nanofluid. By varying the sonication time (1–5 h) to formulate 0.5 vol% Al_2_O_3_/W nanofluid, the TC was monitored [126]. The sonication time increment was found to enhance TC as nanoparticles sedimentation reduced.

#### 4.8.2. Hybrid Nanofluids

The impact of sonication time (2.5–10 min) on the TC of DIW-based Fe_3_O_4_ (0.494–2.428 wt%) + CNT (0.105–1.535 wt%) nanofluids was investigated [127]. With sonication time of 5 min, peak TC of 15.59% (0.494% Fe_3_O_4_ + 0.105% CNT) and 34.26% (2.428% Fe_3_O_4_ + 1.535% CNT) was reported at 55 °C. The relationship between sonication time (0.5–4 h) and TC of DW-based Ag-γAl_2_O_3_ NFs (0.005–0.1 vol%) was examined [114]. At sonication time of 2 h, peak TC was achieved using the nanofluid with a mixing ratio of 50:50 after which reduction is noticed. The influence of varying SDS mass percent, sonication time (20–150 min), mixing ratio, and concentration on the TC of DW-based nanofluids (CuO–MgO–TiO_2_) was investigated [116]. Maximum TC was reached with increasing concentration when the nanofluid with a mixing ratio of 60:30:10 was sonicated for 140 min. Variation in the SDS weight concentration, mixing ratio, sonication time (20–80 min), and volume concentration of MgO–TiO_2_/DW nanofluids was examined. TC was augmented with increasing concentration using the nanofluid with a mixing ratio of 80:20 and 0.35 wt% of SDS, and sonicating the same for 75 min.

## 5. TC Enhancement Techniques

### 5.1. Magnetic Field

Some nanoparticles used in the formulation of nanofluids have magnetic properties that distinct them from other nanoparticles. Magnetic nanofluids are formed by the suspension of ferromagnetic nanoparticles in different base fluids. Depending on the nature of magnetic nanoparticles, magnetic field strength and direction, and type of magnetic field source, particles of magnetic nanofluid form diverse arrangements of chain clusters which tend to affect the thermal properties. The alignment of the particles to form chains on exposure to the magnetic field has been reported to be a veritable tool for the manipulation of the TC of nanofluids [128]. Tuning of the magnetic field has found potential applications in sealing, heat transfer, sensors, nuclear and solar systems, ink jet printers, biomedical, loud speakers, dampers, etc. [128]. The influence of magnetic field strength on nanofluid TC is presented in Figure 9.

Studies on the deployment of the magnetic field to augment the TC of magnetic nanofluids were pioneered using DIW-based Fe and Fe_3_O_4_ nanofluids with varying volume fractions and under increasing and different orientations of magnetic field [130]. The TC of water and n-decane based γ-Fe_2_O_3_ and CoFe_2_O_4_ nanofluids at 25 °C and exposed to an external magnetic field was examined. A reduction in the TC of the nanofluids (40% by CoFe_2_O_4_ and 50% by γ-Fe_2_O_3_) as the magnetic field intensity increased up to 30 mT was observed [131]. At ambient temperature, the TC of Fe_3_O_4_/kerosene nanofluids with varying volume fractions (0.031–7.8%) and under different (parallel and perpendicular) and increasing magnetic field (0–500 G) was accessed [132]. Peak enhancement of 300% for 6.3% concentration when the magnetic field of 82 G was positioned parallel to the temperature gradient was recorded. Exposure of the nanofluid to higher magnetic field strength resulted in TC depreciation. However, no appreciable enhancement was observed when the magnetic field was positioned perpendicular to the temperature gradient. The TC improvement and trend agreed with the results published using a similar nanofluid exposed to the same strength and orientation of magnetic field except that the peak enhancement occurred using volume fraction of 0.078% [93].

The claim that nanofluid TC was enhanced by exposing a magnetic field parallel to the direction of the temperature gradient is also supported by [53], and the existence of peak TC enhancement at a certain magnetic field strength is corroborated by [54]. The parallel arrangement of the magnetic field to the direction of the temperature gradient aids energy transport in the nanofluid as the formed chain structure aligns with the magnetic field direction to quicken energy transportation process [133].

On exposing Fe_3_O_4_/kerosene nanofluids with varying volume fraction (1.12–4.7%) and temperature (25–65 °C) to increasing magnetic field strengths (0–1200 G) and different magnetic field directions, the TC was investigated [133]. Increasing concentration and magnetic field strength (up to 885 G) were observed to enhance the TC with a peak enhancement of 30% when the magnetic field was applied parallel to the temperature gradient direction. The enhancement recorded was attributed to the formation of a zipper-like structure which was reversible. However, temperature increase was noticed to reduce the TC on exposure to increasing magnetic field magnitude. This was in agreement with a later study on the impact of temperature (20–60 °C), volume fraction (0.25–4.8%), and magnetic field (0.021–0.145 T) on the TC of water-based Fe_2_O_3_ and Fe_3_O_4_ nanofluids [134]. Increasing magnetic field strength and volume fraction enhanced Fe_2_O_3_ and Fe_3_O_4_ nanofluids by 15–38.5% and 13–176%, respectively, but temperature rise detracted it. However, an improvement in the TC of Fe_3_O_4_/glycerol nanofluids with temperature (20–40 °C), volume fraction (0.5–3%), and magnetic field strength (120–600 G) rise was reported [55] which contradicted the findings of reported in previous studies [115,133,134]. A maximum enhancement of 16.9% was observed.

An investigation of the effect of utilizing a constant and oscillating magnetic field on the TC of water-based 1 vol% Fe_3_O_4_ and 2 vol% CNT–Fe_3_O_4_ nanofluids was conducted [135]. The use of a constant magnetic field revealed an up-and-down trend with an increase in magnetic field strength over time which was due to the alignment of chainlike structure leading to increased TC, and the thickening of chains and settling causing TC reduction. However, engaging an oscillating magnet field showed an increment in TC as the magnetic field intensified with time. At magnetic field strength of 700 G, average TC was improved by 24.3% and 22.6% for CNT–Fe_3_O_4_ and Fe_3_O_4_ nanofluids, respectively, when the influence of an oscillating magnetic field was compared to that of a constant magnetic field. Similarly, an alternating magnetic field was found to be better than a constant magnetic field for the augmentation of the TC of NiO/DIW nanofluids under increasing volume fraction, temperature, and magnetic field strength [136]. This was because imposing an alternating magnetic field intensifies the velocity and randomness of nanoparticles leading to TC improvement while the use of a constant magnetic field causes nanoparticle chain formation leading to the magnification of TC. The impact of volume fraction and magnetic field strength was observed to be significant.

The influence of duration (0–8 min), magnetic field strength (0.1–0.2 T), and mass concentration (0.45–1.35 wt%) on the TC of Fe_3_O_4_, CuO, and Fe_3_O_4_–CuO nanofluids was examined [129]. Increasing concentration and magnetic field strength were found to augment the nanofluid TC while increasing duration reduced it. The TC of water-based Fe_3_O_4_ and Fe_3_O_4_ + CNT nanofluids under increasing duration (0–60 min), magnetic field strength (330–700 mT), temperature (25–35 °C), and volume fraction (0.45–1.35%) was accessed [115]. The TC was improved as magnetic field strength (up to 470 mT) and volume fraction increased as duration and temperature reduced. Peak enhancement was 151.3% for 0.9% Fe_3_O_4_ + 1.35% CNT nanofluid.

The above studies revealed that there exist contradictory results concerning the influence of temperature rise on the TC of nanofluids when exposed to increasing magnetic field intensity, which needs to be further investigated to have a better view of this observation and to have a well-informed understanding of the physics and mechanism behind this. Additionally, maximum TC attained at a certain magnetic field strength is observed to be a function of nanoparticle type, base fluid type, nanoparticle concentration, temperature, magnetic field strength, and magnetic field type.

### 5.2. Electric Field

Very limited studies have been performed concerning the influence of electric fields on the TC of nanofluids. In a pioneering work, the impact of temperature (26.6–90 °C) and electric field (0–1000 V/mm) intensity on the TC of 30 vol% Al_2_O_3_/silicone oil nanofluid was investigated [137]. Exposure of the nanofluid at 26.6 °C to an increasing electric field from 0 V/mm to 700 V/mm showed a slight increase in the TC from 0.2454 W/m K to 0.2916 W/m K, which surged by 48% on increasing the electric field to 800 V/mm. The TC remained unchanged with a further rise in the electric field. An increment in the temperature of the nanofluid under exposure to an increasing electric field revealed a reduction in the TC. The effect of nanoparticle size (20 nm and 50 nm), temperature (15–55 °C), concentration (0.1–1.5 wt%), and electric field (0–1.2 MV/m) on the TC of αAl_2_O_3_/transformer oil nanofluids [138]. Increasing temperature, concentration, and electric field were observed to augment the nanofluid TC while increment in nanoparticle size has an insignificant effect on the TC. The TC recorded is strongly related to the Brownian motion phenomenon. A contradiction regarding the effect of temperature on the TC of nanofluids exposed to an electric field is observed which calls for further studies on this concern in addition to the scarcity of literature in the public domain. The effect of electric field intensity on nanofluid TC is illustrated in Figure 10.

## 6. Green Nanofluids

Though with very limited studies, the TC of green base fluid and nanofluid has also been investigated. At 2 vol% and 80 °C, the TC of Al_2_O_3_/40 biogylcol-60 water and Al_2_O_3_/60 biogylcol-40 water nanofluids was found to be enhanced by 24% and 13%, respectively [67]. The higher TC of water was suggested to have influenced the obtained results. By investigating the TC of bioglycol-based Al_2_O_3_ nanofluids, a maximum enhancement of 17% was observed at 30 °C while peak TC was recorded at 70 °C, all at a concentration of 1 vol%, despite subjecting the nanofluids to increasing temperature from 30 °C to 80 °C [51]. Additionally, the use of the green base fluid was observed to result in a higher enhancement of the nanofluid TC compared to those of EG (9%) and PG (3.6%). A green nanofluid formulated by the suspension of TiO_2_–SiO_2_ (20:80) nanoparticles into bio-glycol–water (60:40) was examined for TC under changing temperature (30–70 °C) and concentration (0.5–3 vol%) [52]. Both the concentration and temperature increase were observed to augment nanofluid TC with peak improvement of 12.5% for 3 vol% at 70 °C.

Similarly, the impact of varying temperature (30–70 °C) and concentration (0.5–2.5 vol%) on the TC of TiO_2_–SiO_2_ (20:80)/bio-glycol–water (40:60) nanofluids was accessed [139]. TC enhancement of 0.7–11.2% was recorded as temperature and concentration rise directly affected it. The TC of green nanofluids formulated by suspending eco-friendly ZnO in green glycerol at varying mass concentrations (0.01–1 wt%) was found to enhance with increasing concentration [28]. The influence of mixing ratio (70:30, 50:50, 30:70, and 10:90) and temperature (30–70 °C) on the TC of 0.5 vol% Al_2_O_3_–SiO_2_/water-green glycol (60:40) was investigated [47]. Peak TC of 21.2% was achieved using Al_2_O_3_–SiO_2_ nanofluid with a mixing ratio of 30:70 and at 70 °C. With an EG-based hybrid nanofluid formulated using fruit bunch and GO nanoparticles, TC was recorded to be improved by 6.47% (at 0.06 wt% and 40 °C) [50]. These studies are strong indicators for the development of green nanofluids and their applications. TC of green nanofluid is presented in Figure 11.

## 7. Future Perspective and Challenges

Different classifications of nanoparticles, base fluids, and surfactants have been deployed as materials for the formulation of nanofluids. The advantages of metal oxide nanofluids over other classes of nanofluids have attracted the attention of the nanofluid research community. TC is considered to be the foremost thermal property of nanofluids as this relates to different applications of nanofluids. Therefore, the TC of metal oxide nanofluids is crucial to the future of nanofluid research, which has prompted this present review work. The reported inconsistency in the TC measurement of metal oxide nanofluids needs to be addressed through the development of TC devices using various TC measurement techniques. Improved development of transient hot wire and thermal constants analyzer techniques is very important to correct the disparity in TC results for different nanofluids. Sensitivity Stability remains a critical factor in nanofluid research. Nanofluid stability is directly related to the TC of metal oxide nanofluids. Stable nanofluids are to be formulated by optimizing the various preparation characteristics [32] to improve their TC values and nanofluid applications. In addition, the functionalization of metal oxide nanoparticles in the formulation of nanofluids can be considered as an option in the future to enhance their stability and augment TC values.

Moderate studies have been conducted on the impact of magnetic field on the TC of metal oxide. However, very limited ones have experimented the effect of alignment on metal oxide nanofluid TC. More works need to be conducted to investigate the alignment of different nanofluids (mono, hybrid, magnetic, and non-magnetic) under the influence of magnetic field with different orientation and intensity. The effect of electric field on nanoparticle alignment in nanofluid and TC of nanofluids is very scare in the open literature and further studies in this respect are expected in the future. In addition, studies are to be intensified concerning the effect of electric field intensity and orientation on the TC of nanofluids. In terms of sustainability and eco-friendly environment, investigation on less toxic green base fluids, nanoparticles, nanofluids, and synthetic routes marked with increased TC is expected to increase shortly as studies are presently very limited [28,47,140,141,142].

## 8. Conclusions

A review of the TC measurement and enhancement of metal oxide nanofluids has been conducted. Generally, the suspension of nanoparticles of different metal oxides in diverse base fluids has been observed to enhance the TC of the base fluids, even regarding the use of hybrid nanoparticles (with one metal oxide or both metal oxides). The TC and the resultant enhancement were noticed to be dependent on several contributing factors, such as temperature, nanoparticle and base fluid characteristics, measurement technique, alignment, concentration, sonication characteristics, surfactant presence, type, and concentration with key ones reviewed and discussed in this present work. The transient hot wire is the most used TC measuring technique with the issue of nanofluid stability being critical to its usage and accuracy. The concentration of nanofluids has a direct influence on the TC while nanoparticle size, temperature, mixing ratio, and sonication characteristics have conflicting effects on it. The deployment of electric and magnetic fields with increasing intensity and concentration was found to augment the TC of metal oxide nanofluids, but the influence of increasing temperature was marked by controversial results. Undoubtedly, electric and magnetic fields can be utilized to control nanofluid TC with the former requiring intense future studies. Brownian motion, nanoparticle clustering, nanolayer, and Ballistic nature remained the most important mechanisms responsible for the uncharacteristic TC enhancement of nanofluids. The development of green synthetic processes, base fluid, nanoparticles, nanofluid, and hybrid nanofluid is envisioned to be critical to the future of nanofluid research and enhancement of TC for thermal transport applications.

## Figures and Tables

**Figure 1 nanomaterials-13-00597-f001:**
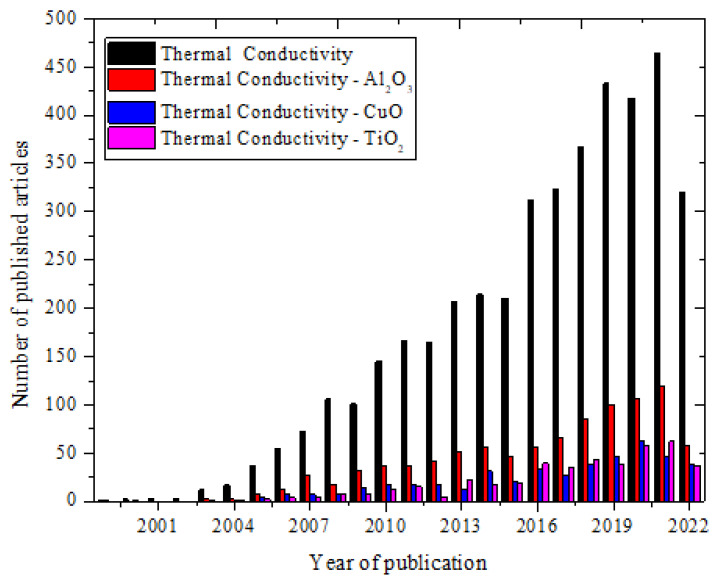
Temporal trend of thermal conductivity of nanofluids, and top metal oxide nanofluids of Al_2_O_3_, CuO, and TiO_2_ (Sourced from Scopus database, 31 August 2022).

**Figure 2 nanomaterials-13-00597-f002:**
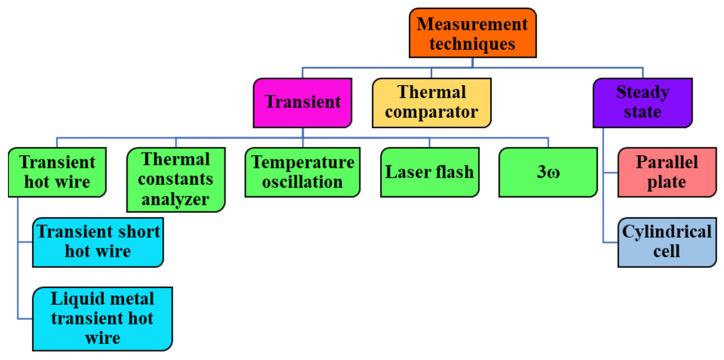
Division and sub-division of thermal conductivity measurement techniques.

**Figure 3 nanomaterials-13-00597-f003:**
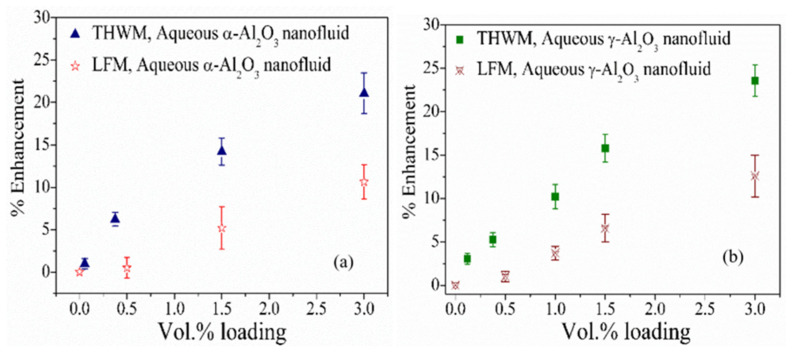
Thermal conductivity enhancement of (**a**) α-Al_2_O_3_ and (**b**) γ-Al_2_O_3_ nanofluids under increasing concentration using different measurement techniques [73].

**Figure 4 nanomaterials-13-00597-f004:**
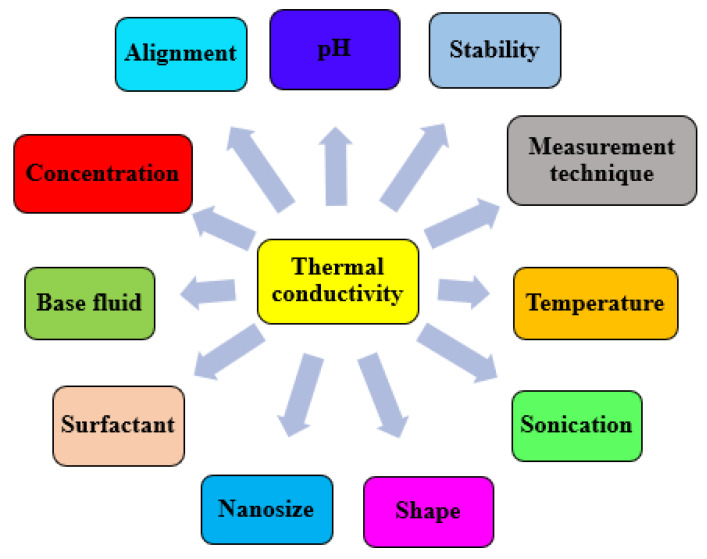
Factors affecting thermal conductivity of nanofluids.

**Figure 5 nanomaterials-13-00597-f005:**
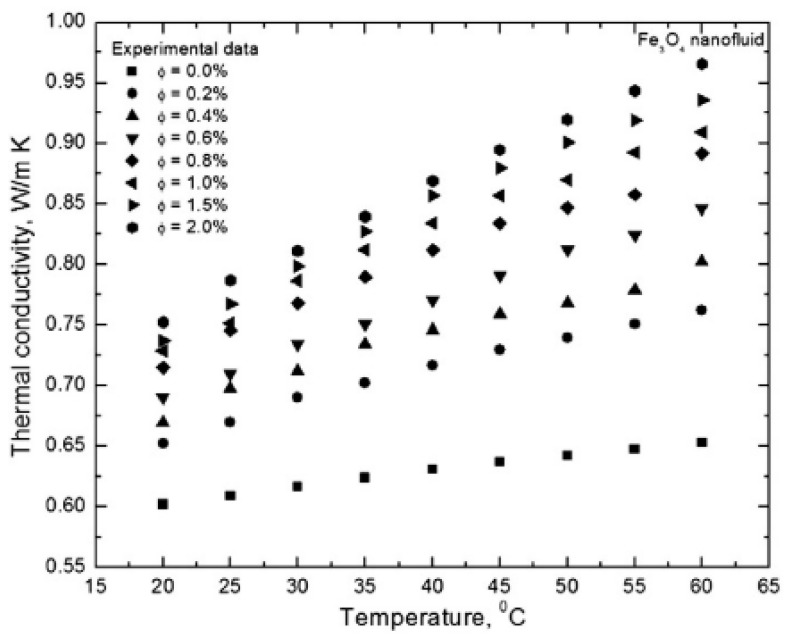
Effect of increasing temperature on thermal conductivity of Fe_3_O_4_/water nanofluids under different volume concentration [64].

**Figure 6 nanomaterials-13-00597-f006:**
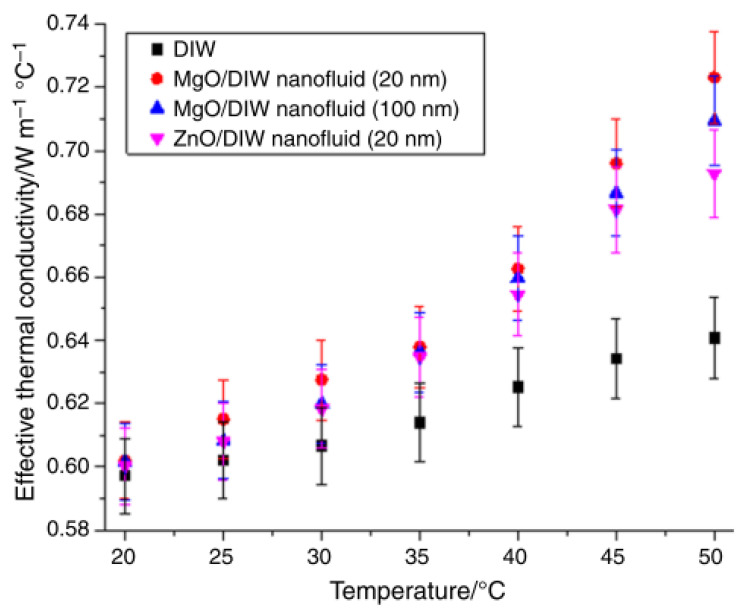
Effect of increasing temperature on effective thermal conductivity of nanofluids with different nanoparticle size [43].

**Figure 7 nanomaterials-13-00597-f007:**
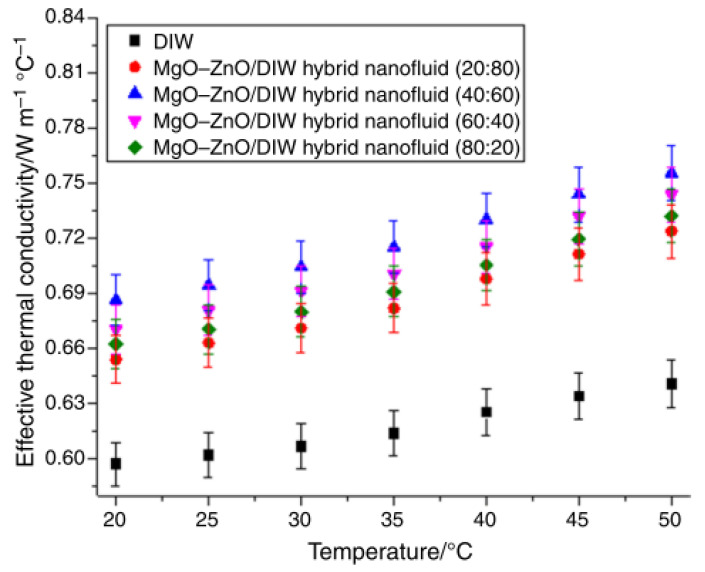
Effect of increasing temperature on effective thermal conductivity of 0.1 vol% MgO–ZnO/DIW nanofluid under varying mixing ratio [43].

**Figure 9 nanomaterials-13-00597-f009:**
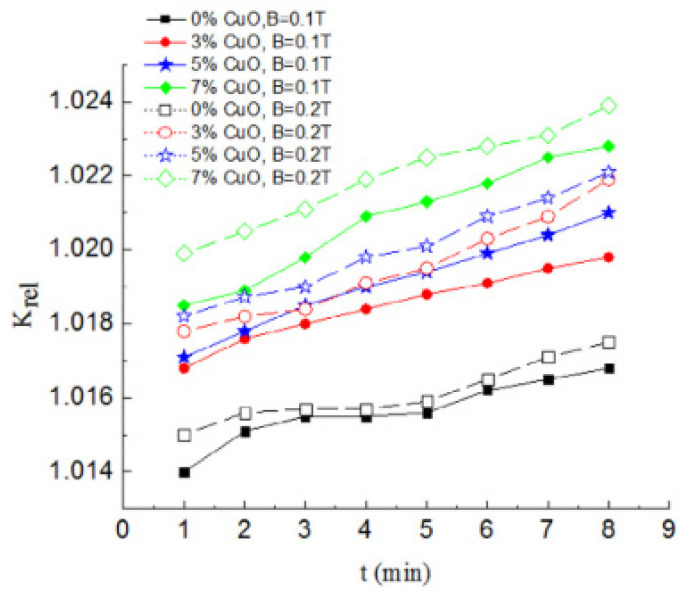
Effect of increasing magnetic field intensity and CuO + 3 wt% Fe_3_O_4_ nanofluids on relative thermal conductivity under increasing exposure duration [129].

**Figure 10 nanomaterials-13-00597-f010:**
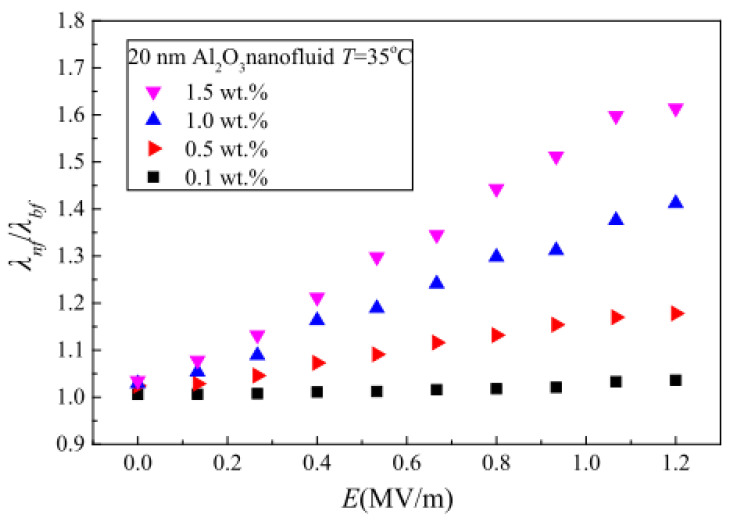
Effect of increasing electric field intensity and concentration on thermal conductivity of 20 nm—Al_2_O_3_ nanofluid at 35 °C [138].

**Figure 11 nanomaterials-13-00597-f011:**
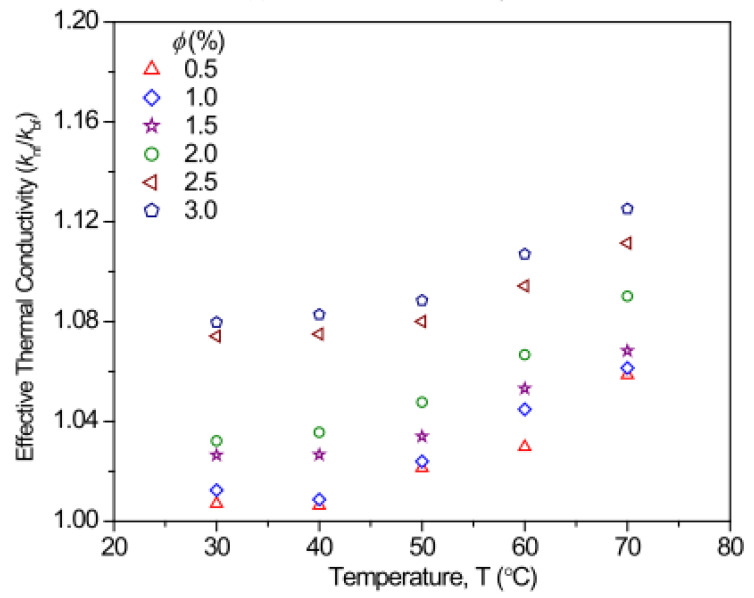
Effect of increasing temperature on effective thermal conductivity of TiO_2_–SiO_2_ (20:80)/biogycol-DW (40:60) nanofluids under different volume concentration [52].

**Table 1 nanomaterials-13-00597-t001:** Thermal conductivity and measuring techniques of different mono and hybrid nanofluids with dissimilar particle sizes at varying concentrations and temperatures.

References	Nanoparticles (Mixing Ratio)	Base Fluid	Concentration (vol%)	Size (nm)	Temperature (°C)	Measuring Techniques	Remarks
[56]	ND-Fe_3_O_4_ (72%:28%)	DW and DW-EG mixtures	0.05–0.2	21.24	20–60	THW	Maximum enhancement of 13.4–17.8%.
[57]	ND-Co_3_O_4_ (67%:33%)	DW	0.05–0.5 wt.%	-	20–60	THW	Maximum enhancement of 9.0–16.0%.
[58]	MWCNT-Al_2_O_3_ (1:1)	DIW	0.1	-	Room	THW	Peak enhancement of 20.68%.
[59]	TiO_2_–SiO_2_ (40:60 vol%)	EG	0.5–3.0	TiO_2_ (50) and SiO_2_ (30)	30–70	THW	Peak enhancement of 22.1%.
[60]	Al_2_O_3_-Cu (90:10 wt.%)	DIW	0.1–2.0	17	Room temp.	THW	Peak enhancement of 1.47–12.11%.
[45]	MWCNT-Fe_2_O_3_ (0.05:0.02 wt.%)	W	MW-0.05 wt.%; Fe_2_O_3_-0.01–0.16 wt.%	-	Room temp.	THW	Peak enhancement of 27.75%.
[61]	TiO_2_, Al_2_O_3_	DIW and EG	1–5 vol%	15 (TiO_2_) 80 (Al_2_O_3_)	20–60	THW	18% (TiO_2_/EG) and 12% (Al_2_O_3_/EG)
[62]	CuO and Al_2_O_3_	DW	1–4 vol%	38.4 (Al_2_O_3_), 28.6 (CuO)	21–51	Temperature oscillation	36% (CuO) and 24.3% (Al_2_O_3_)
[63]	Al_2_O_3_	DIW	0.01–0.3%	30	Room temp.	THW	Peak enhancement of 1.44%
[64]	Fe_3_O_4_	DIW	0–2 vol%	13	20–60	THW	Improvement of 48% (maximum).
[65]	Fe_3_O_4_	kerosene	0–1 vol%	15	10–60	THW	Enhancement of 34% (maximum)
[66]	SiO_2_	EG–water (0–100)	0.3% (mass)	30	25–45	THW	Reduction as EG content increased.
[67]	Al_2_O_3_	Bioglycol–water (40:60 and 60:40)	0.5–2 vol%	13	30–80	THW	Bioglycol–water (40:60) has higher value of thermal conductivity.
[68]	TiO_2_	EG	0–7 vol%	5	10–50	THW	Enhancement of 19.5% (maximum).
[69]	Al_2_O_3_ and CuO	EG, pump oil, DIW, and EO	0–8%	Al_2_O_3_ (28) and CuO (23)	Room temp.	Steady state parallel plate	40% and 12% for Al_2_O_3_/EG and Al_2_O_3_/water, respectively.
[70]	Al_2_O_3_	DIW	0.1–2.5%	13 and 20	Room temp.	Temperature oscillation	13 nm (lower) and 2.5% (highest) resulted in peak thermal conductivity.

## Data Availability

All data is available in the manuscript.

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
