# Peer review of "Thermal Conductivity Enhancement of Metal Oxide Nanofluids: A Critical Review"

_nanomaterials, 2023, doi:10.3390/nano13030597_

Round 1

Reviewer 1 Report

The main observations are listed below. The acceptance of the manuscript would depend on the revision. The author needs to provide a point-by-point response or provide a rebuttal.

1. The caption of Table 1 does not reflect the entire features. Rectify it.

2. The division of the manuscript should be stated in the last part of the introduction section.

3. Lines 149 to 150 should be cross-checked for correctness.

4. Lines 454 to 457 with (XXXX) should be addressed.

5. Figures should be placed close to the point of their first mention.

6. Figures’ captions need to be fixed.

Reviewer 2 Report

This study provides a comprehensive review of the thermal conductivity enhancement of metal oxide nanofluids. The structure is basically well-prepared and organized; however, this review paper still should be improved. Please see the comments below:

1.     The Abstract should contain answers to the following questions: What problem was studied and why is it important? What methods were used? What are the important results? What conclusions can be drawn from the results? What is the novelty of the work and where does it go beyond previous efforts in the literature? Add the main findings and objective of the current study in the abstract.

2.     Introduction: There are many review papers related to this topic. The novelty of the paper needs to be justified and clearly defined in the last paragraph of the introduction. It includes the clear difference between the available literature and previous works. The authors are asked to provide the limitations of the previous correlated works and then linked those limitations to the current ideas and contributions of the current work.

3.     Please avoid too many lumped references.

4.     Future perspectives and challenges of the technology should be provided before the conclusion.

5.     References: The papers published in the current year and the last three years should be added.

Reviewer 3 Report

Nanofluids are interesting but challenging. Authors tried to review thermal conductivity enhancement of metal oxide nanofluids. It should be encouraged.

Few comments to strengthen the manuscript:

1. Beyond the factors authors mentioned such as particle size, particle clustering, magnetic field, there are several other factors that influence the thermal conductivity such as pH, solvent, alignment, etc. It is suggested to add more contents.  

2. References are not comprehensive. Suggest adding the following:

Enhanced thermal conductivity by aggregation in heat transfer nanofluids containing metal oxide nanoparticles and carbon nanotubes

J Wensel, B Wright, D Thomas, W Douglas, B Mannhalter, W Cross, ...

Applied Physics Letters 92 (2), 023110, 2008

Effects of  on heat transfer nanofluids containing  and  nanoparticles

CT Wamkam, MK Opoku, H Hong, P Smith

Journal of Applied Physics 109 (2), 024305, 2011

Effects of alignment, pH, surfactant, and solvent on heat transfer nanofluids containing Fe2O3 and CuO nanoparticles

H Younes, G Christensen, X Luan, H Hong, P Smith

Journal of Applied Physics 111 (6), 064308, 2012

Alignment of carbon nanotubes comprising magnetically sensitive metal oxides in heat transfer nanofluids

H Hong, X Luan, M Horton, C Li, GP Peterson

Thermochimica Acta 525 (1-2), 87-92, 2011

Thermal conductivity of nanofluids

H Younes, G Christensen, D Li, H Hong, AA Ghaferi

Journal of Nanofluids 4 (2), 107-132, 2015

Effects of solvent hydrogen bonding, viscosity, and polarity on the dispersion and alignment of nanofluids containing Fe2O3 nanoparticles

G Christensen, H Younes, H Hong, P Smith

Journal of Applied Physics 118 (21), 214302, 2015

3. Some grammar and spelling mistakes in the manuscript.

Round 2

Reviewer 1 Report

This revised version is made elaborately. The paper can be accepted now.

Reviewer 2 Report

The authors have addressed all the comments.